# A Dataset of Scalp EEG Recordings of Alzheimer's Disease, Frontotemporal Dementia and Healthy Subjects from Routine EEG

Andreas Miltiadous [1], Katerina D. Tzimourta [1,2], Theodora Afrantou [3], Panagiotis Ioannidis [3], Nikolaos Grigoriadis [3], Dimitrios G. Tsalikakis [2], Pantelis Angelidis [2], Markos G. Tsipouras [2], Euripidis Glavas [1], Nikolaos Giannakeas [1] and Alexandros T. Tzallas [1,*]

1   Department of Informatics and Telecommunications, University of Ioannina, 47100 Arta, Greece; a.miltiadous@uoi.gr (A.M.); ktzimourta@uowm.gr (K.D.T.); eglavas@uoi.gr (E.G.); giannakeas@uoi.gr (N.G.)
2   Department of Electrical and Computer Engineering, University of Western Macedonia, 50100 Kozani, Greece; dtsalikakis@uowm.gr (D.G.T.); paggelidis@uowm.gr (P.A.); mtsipouras@uowm.gr (M.G.T.)
3   2nd Department of Neurology, AHEPA University Hospital, Aristotle University of Thessaloniki, 54636 Thessaloniki, Greece; afrantou@gmail.com (T.A.); ioannidispanosgr@yahoo.gr (P.I.); ngrigoriadis@auth.gr (N.G.)
*   Correspondence: tzallas@uoi.gr

**Abstract:** Recently, there has been a growing research interest in utilizing the electroencephalogram (EEG) as a non-invasive diagnostic tool for neurodegenerative diseases. This article provides a detailed description of a resting-state EEG dataset of individuals with Alzheimer's disease and frontotemporal dementia, and healthy controls. The dataset was collected using a clinical EEG system with 19 scalp electrodes while participants were in a resting state with their eyes closed. The data collection process included rigorous quality control measures to ensure data accuracy and consistency. The dataset contains recordings of 36 Alzheimer's patients, 23 frontotemporal dementia patients, and 29 healthy age-matched subjects. For each subject, the Mini-Mental State Examination score is reported. A monopolar montage was used to collect the signals. A raw and preprocessed EEG is included in the standard BIDS format. For the preprocessed signals, established methods such as artifact subspace reconstruction and an independent component analysis have been employed for denoising. The dataset has significant reuse potential since Alzheimer's EEG Machine Learning studies are increasing in popularity and there is a lack of publicly available EEG datasets. The resting-state EEG data can be used to explore alterations in brain activity and connectivity in these conditions, and to develop new diagnostic and treatment approaches. Additionally, the dataset can be used to compare EEG characteristics between different types of dementia, which could provide insights into the underlying mechanisms of these conditions.

**Dataset:** 10.18112/openneuro.ds004504.v1.0.2.

**Dataset License:** CC0

**Keywords:** electroencephalography; routine EEG; Alzheimer's disease; frontotemporal dementia; resting state

## 1. Summary

Alzheimer's disease (AD) and frontotemporal dementia (FTD) are both progressive neurodegenerative disorders that affect the elderly [1]. AD is the most frequently diagnosed dementia type, accounting for 60–80% of cases, while FTD is relatively rare, accounting for 5–10% of cases [2]. Both neurological conditions are characterized by cognitive decline and behavioral changes, and affect the brain in different ways, resulting in distinct (yet possibly overlapping) symptoms [3]. An initial AD sign is difficulty in recalling events

related to short-term memory and it progresses to speech and orientation difficulties, lack of self-care, or behavioral alterations. The initial sign of frontotemporal dementia (FTD) can vary depending on which part of the brain is affected first. However, behavioral changes and personality changes are often the initial symptoms in the behavioral variant of FTD, which is the most common variant of the disease [3]. Currently, there is no cure for either condition, while available treatments provide limited symptomatic relief [4].

A combination of clinical evaluation, neurological testing, and neuropsychological testing is used to make the diagnosis of Alzheimer's disease and frontotemporal dementia. These disorders may also be diagnosed with the use of imaging tests such as positron emission tomography (PET) [5] or magnetic resonance imaging (MRI) [6]. The early symptoms of both frontotemporal dementia and Alzheimer's disease might be mild and overlap with other neurodegenerative illnesses or mental problems, making a diagnosis difficult in both cases. Better detection technologies are thus required in order to help in the early identification of these illnesses. A timely diagnosis is essential because early care can help postpone the emergence of symptoms that grow more severe and enhance quality of life [7]. A neurodegenerative condition can be challenging to live with, but an early diagnosis enables the installation of safety precautions, legal and financial planning, and emotional support services, which can help people and their families manage. Therefore, there is an urgent need for new detection methods that can help with the early detection of frontotemporal dementia and Alzheimer's disease, which can eventually lead to better outcomes for those who have these disorders.

Electroencephalography (EEG) has become a potential method for the identification and monitoring of Alzheimer's disease and frontotemporal dementia, in addition to clinical evaluation and imaging testing [8]. EEG measures brain electrical activity and can identify anomalies in brain waves linked to certain disorders [9–12]. Then, using machine learning techniques, these signals can be automatically analyzed to find patterns that might point to sickness. For instance, machine learning models can spot a slowing of brain waves in specific areas of the brain in AD, and they can spot alterations in connectivity between distinct brain regions in FTD [2]. The automatic identification of these illnesses using machine learning and EEG readings is still in its infancy and needs more study and validation. However, the promise of EEG and machine learning as non-invasive, affordable, and accessible detection methods emphasizes the necessity of ongoing study in this area, which may ultimately result in the improved and more prompt diagnosis of Alzheimer's disease and frontotemporal dementia.

The aim of this study was to collect the electrical activity of the brain of elderly patients with AD and FTD, and healthy age-matching controls, during the eye resting state using EEG. These recordings are structured in the Brain Imaging Data Structure (BIDS) format, which is a standardized format for organizing and describing neuroimaging data [13]. BIDS was developed to improve the consistency, compatibility, and ease of use of neuroimaging data across different research groups and institutions. Researchers focusing on these neurodegenerative disorders will find the released dataset of EEG recordings from people with Alzheimer's disease and frontotemporal dementia, and healthy controls (CN), to be an essential tool. Researchers will be able to examine the diseases' underlying mechanisms, find potential biomarkers for early identification, and test new treatments thanks to the dataset. The development and testing of machine learning methodologies, which can be used to automatically detect and categorize diseases based on EEG signals, require the availability of datasets like this. With the use of this dataset, researchers may test and refine their algorithms, advancing the area of machine learning-based neurodegenerative disease diagnosis. Overall, this dataset has the potential to significantly advance our understanding of Alzheimer's disease, frontotemporal dementia, and the role of EEG in their diagnosis and management.

As a result of this work, EEG recordings from 88 subjects have been registered and cleared of artifacts and have been made available to the cognitive neuroscience research community. In total, 36 of them were diagnosed with AD, 23 with FTD, and 29 were CN. Prior to the publication of this dataset, two studies regarding machine learning methodologies for the classification or severity quantification of AD and FTD have been published, using a subset of participants [2,14].

## 2. Data Description

This dataset contains the EEG resting state-closed eyes recordings from 88 subjects in total. A total of 36 of them were diagnosed with Alzheimer's disease (AD group), 23 were diagnosed with frontotemporal dementia (FTD group), and 29 were CN. The cognitive and neuropsychological state was evaluated by the international Mini-Mental State Examination (MMSE) [15]. The MMSE score ranges from 0 to 30, with a lower MMSE indicating more severe cognitive decline.

### 2.1. EEG Recordings

Recordings include the EEG signal from 19 scalp electrodes (Fp1, Fp2, F7, F3, Fz, F4, F8, T3, C3, Cz, C4, T4, T5, P3, Pz, P4, T6, O1, and O2) and 2 reference electrodes, placed according to the 10–20 international system. The sampling rate was 500 Hz and the resolution was 10 uV/mm. Each recording lasted approximately 13.5 min for the AD group (min = 5.1, max = 21.3), 12 min for the FTD group (min = 7.9, max = 16.9), and 13.8 min for the CN group (min = 12.5, max = 16.5). In total, 485.5 min of AD, 276.5 min of FTD, and 402 min of CN recordings were collected and are included in the dataset.

### 2.2. Participants

All the recordings were acquired from routine EEG of patients of the aforementioned groups. The duration of the disease was measured in months and the median value was 25 with the IQR range (Q1-Q3) being 24–28.5 months. Concerning the AD group, no dementia-related comorbidities have been reported. The initial diagnosis for the AD and FTD patients was performed according to the criteria provided by the Diagnostic and Statistical Manual of Mental Disorders, 3rd ed., revised (DSM-IIIR, DSM IV, ICD-10) [16] and the National Institute of Neurological, Communicative Disorders and Stroke—Alzheimer's Disease and Related Disorders Association (NINCDS—ADRDA) [17]. The average MMSE for the AD group was 17.75 (SD = 4.5), for the FTD group it was 22.17 (SD = 8.22), and for the CN group it was 30. The mean age of the AD group was 66.4 years (SD = 7.9), for the FTD group it was 63.6 (SD = 8.2), and for the CN group it was 67.9 (SD = 5.4). Table 1 presents a detailed description of each participant. Participants have been anonymized and personal information has not been disclosed, following GDPR restrictions.

**Table 1.** Participant Description. In the Group column, A indicates AD patient, F indicates FTD patient, and C indicates a healthy subject. In the Gender column, F indicates female and M indicates male.

| Participant_id | Gender | Age | Group | MMSE |
|---|---|---|---|---|
| sub-001 | F | 57 | A | 16 |
| sub-002 | F | 78 | A | 22 |
| sub-003 | M | 70 | A | 14 |
| sub-004 | F | 67 | A | 20 |
| sub-005 | M | 70 | A | 22 |
| sub-006 | F | 61 | A | 14 |
| sub-007 | F | 79 | A | 20 |
| sub-008 | M | 62 | A | 16 |
| sub-009 | F | 77 | A | 23 |
| sub-010 | M | 69 | A | 20 |

**Table 1.** *Cont.*

| Participant_id | Gender | Age | Group | MMSE |
|---|---|---|---|---|
| sub-011 | M | 71 | A | 22 |
| sub-012 | M | 63 | A | 18 |
| sub-013 | F | 64 | A | 20 |
| sub-014 | M | 77 | A | 14 |
| sub-015 | M | 61 | A | 18 |
| sub-016 | F | 68 | A | 14 |
| sub-017 | F | 61 | A | 6 |
| sub-018 | F | 73 | A | 23 |
| sub-019 | F | 62 | A | 14 |
| sub-020 | M | 71 | A | 4 |
| sub-021 | M | 79 | A | 22 |
| sub-022 | F | 68 | A | 20 |
| sub-023 | M | 60 | A | 16 |
| sub-024 | F | 69 | A | 20 |
| sub-025 | F | 79 | A | 20 |
| sub-026 | F | 61 | A | 18 |
| sub-027 | F | 67 | A | 16 |
| sub-028 | M | 49 | A | 20 |
| sub-029 | F | 53 | A | 16 |
| sub-030 | F | 56 | A | 20 |
| sub-031 | F | 67 | A | 22 |
| sub-032 | F | 59 | A | 20 |
| sub-033 | F | 72 | A | 20 |
| sub-034 | F | 75 | A | 18 |
| sub-035 | F | 57 | A | 22 |
| sub-036 | F | 58 | A | 9 |
| sub-037 | M | 57 | C | 30 |
| sub-038 | M | 62 | C | 30 |
| sub-039 | M | 70 | C | 30 |
| sub-040 | M | 61 | C | 30 |
| sub-041 | F | 77 | C | 30 |
| sub-042 | M | 74 | C | 30 |
| sub-043 | M | 72 | C | 30 |
| sub-044 | F | 64 | C | 30 |
| sub-045 | F | 70 | C | 30 |
| sub-046 | M | 63 | C | 30 |
| sub-047 | F | 70 | C | 30 |
| sub-048 | M | 65 | C | 30 |
| sub-049 | F | 62 | C | 30 |
| sub-050 | M | 68 | C | 30 |
| sub-051 | F | 75 | C | 30 |
| sub-052 | F | 73 | C | 30 |
| sub-053 | M | 70 | C | 30 |
| sub-054 | M | 78 | C | 30 |
| sub-055 | M | 67 | C | 30 |
| sub-056 | F | 64 | C | 30 |
| sub-057 | M | 64 | C | 30 |
| sub-058 | M | 62 | C | 30 |
| sub-059 | M | 77 | C | 30 |
| sub-060 | F | 71 | C | 30 |
| sub-061 | F | 63 | C | 30 |
| sub-062 | M | 67 | C | 30 |
| sub-063 | M | 66 | C | 30 |
| sub-064 | M | 66 | C | 30 |

**Table 1.** *Cont.*

| Participant_id | Gender | Age | Group | MMSE |
|---|---|---|---|---|
| sub-065 | F | 71 | C | 30 |
| sub-066 | M | 73 | F | 20 |
| sub-067 | M | 66 | F | 24 |
| sub-068 | M | 78 | F | 25 |
| sub-069 | M | 70 | F | 22 |
| sub-070 | F | 67 | F | 22 |
| sub-071 | M | 62 | F | 20 |
| sub-072 | M | 65 | F | 18 |
| sub-073 | F | 57 | F | 22 |
| sub-074 | F | 53 | F | 20 |
| sub-075 | F | 71 | F | 22 |
| sub-076 | M | 44 | F | 24 |
| sub-077 | M | 61 | F | 22 |
| sub-078 | M | 62 | F | 22 |
| sub-079 | F | 60 | F | 18 |
| sub-080 | F | 71 | F | 20 |
| sub-081 | F | 61 | F | 18 |
| sub-082 | M | 63 | F | 27 |
| sub-083 | F | 68 | F | 20 |
| sub-084 | F | 71 | F | 24 |
| sub-085 | M | 64 | F | 26 |
| sub-086 | M | 49 | F | 26 |
| sub-087 | M | 73 | F | 24 |
| sub-088 | M | 55 | F | 24 |

*2.3. Dataset Structure*

This dataset was preprocessed and formed in its current structure in the Human Computer Interaction Laboratory of the Department of Informatics and Telecommunications, University of Ioannina, Greece. It is structured in the BIDS format. The BIDS format specifies the file organization structure and naming convention for all neuroimaging data, including structural and functional MRI and EEG. It also defines metadata that describe the data, in JSON format, such as subject and session identifiers, acquisition parameters, and task information. Making the dataset BIDS compatible ensures the ease of use for other researchers because open-source software (such as EEGLAB [18]) provide tools for analyzing and processing neuroimaging data of BIDS-compliant datasets. Figure 1 provides a description of the dataset structure.

The dataset consists of the following: (1) The dataset_description.json file, which provides information regarding the authors of the dataset, the acknowledgment of the research project that made this work possible, the DOI, the BIDS version of the dataset, the license under which it is published, and the ethics approval statement. (2) The participants.json file, which contains definitions regarding the attributes of the participants, as shown in Table 1. This metadata file is used by software such as EEGLAB to automatically group and label the EEG recordings to the participants. (3) The participants.tsv file, which is a tab-separated file containing the information of Table 1. (4) A folder system of folders named as sub-0XX. Each folder is associated to one participant-id of the participant table. Additionally, each folder contains three files: (A) A sub-0XX-task_eyesclosed_eeg.json file, which contains all the necessary EEG recording information, such as the placement scheme (10–20), the reference (A1 and A2), the model of the device and amplifier used, the channel count, the sampling frequency, the recording duration, and more. (B) A sub-0XX_task-eyesclosed_channels.tsv file, which provides information about electrode location. (C) A sub-0XX_task-eyesclosed_eeg.set file, which contains the EEG recordings of the participant in a .set format, which is one of the four BIDS-allowed EEG formats (those being the European data format .edf, the BrainVision Core Data Format .vhdr or .eeg, the EEGLAB format .set, and the Biosemi format .bdf). The following two facts should be

noted. First, the .set files contain all the necessary recording information; thus, they can also be accessed in a non-BIDS setting. Second, the sub-0XX_task-eyesclosed_channels.tsv and sub-0XX-task_eyesclosed_eeg.json files are the same for each participant, since the same recording setting has been used (except for the recording duration information, which differs); thus, users do not need to examine all of them. (5) The folder derivatives, which contain subfolders with the same structure described before, with the difference that the EEG recordings are preprocessed.

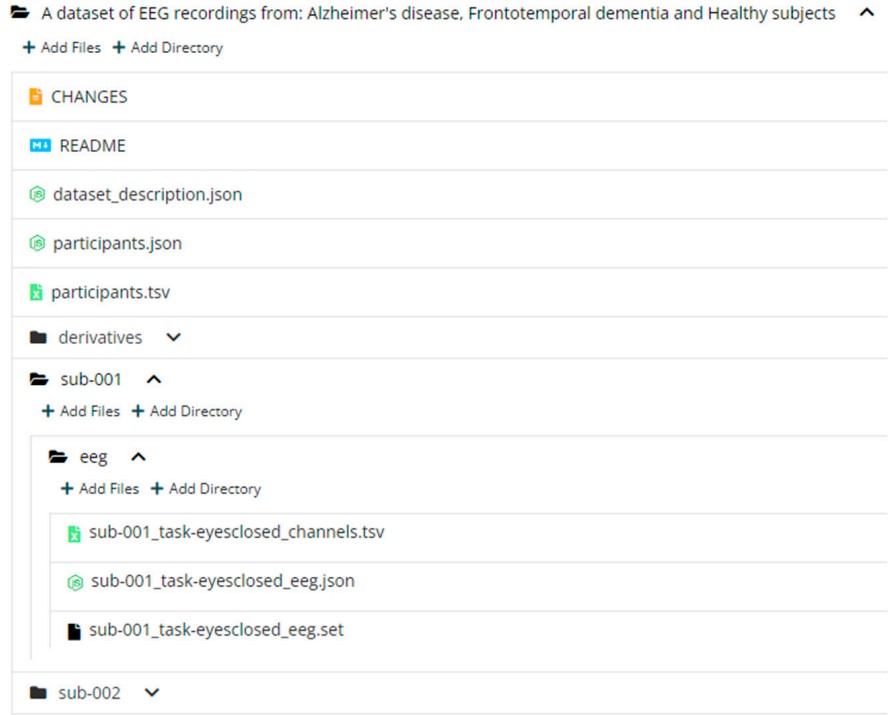

**Figure 1.** The structure of the dataset following the BIDS format.

## 3. Methods

### 3.1. Recording

The recordings of this dataset were collected to investigate functional differences in the EEG activity of AD versus CN, FTD versus CN, and even AD versus FTD. These recordings took place in a clinical routine setting. Recordings were acquired from the 2nd Department of Neurology of AHEPA General Hospital of Thessaloniki by an experienced team of neurologists. A clinical EEG device (Nihon Kohden 2100), with 19 scalp electrodes (Fp1, Fp2, F7, F3, Fz, F4, F8, T3, C3, Cz, C4, T4, T5, P3, Pz, P4, T6, O1, and O2) and 2 electrodes (A1 and A2) placed on the mastoids for an impedance check and as reference electrodes, was used for the recording of the EEG signals. The electrodes were placed according to the 10–20 international system. Each recording was performed according to the clinical protocol with participants being in a sitting position with their eyes closed. The recording montage was referential using Cz for common mode rejection. The sampling rate was 500 Hz and the resolution was 10 uV/mm. This study was approved by the Scientific and Ethics Committee of AHEPA University Hospital, Aristotle University of Thessaloniki, under protocol number 142/12-04-2023. The investigations were carried out following the rules of the Declaration of Helsinki of 1975 (http://www.wma.net/en/30publications/10policies/b3/, accessed on March 2019), revised in 2008.

### 3.2. Preprocessing

Only the derivatives folder, where the preprocessed data is kept, is covered by this section. The following is the EEG signals' preprocessing pipeline. The signals were re-referenced to the average value of A1-A2 after applying a Butterworth band-pass filter with

a frequency range of 0.5 to 45 Hz. The signals were then subjected to the ASR routine, an automatic artifact reject technique that can eliminate persistent or large-amplitude artifacts, which removed bad data periods that exceeded the maximum acceptable 0.5 s window standard deviation of 17 (which is regarded as a conservative window). The ICA method (RunICA algorithm) was then used to convert the 19 EEG signals to 19 ICA components [19]. ICA components categorized as "eye artifacts" or "jaw artifacts" by the EEGLAB platform's automatic classification method "ICLabel" were automatically excluded. It should be mentioned that, even though the recording was done in a resting state with the eyes closed, eye movement artifacts were still identified in certain EEG recordings. Figure 2 represents a snapshot of the same signal in raw form, and in preprocessed form. It can be observed that severe high frequency artifacts have been removed and baseline correction has been applied.

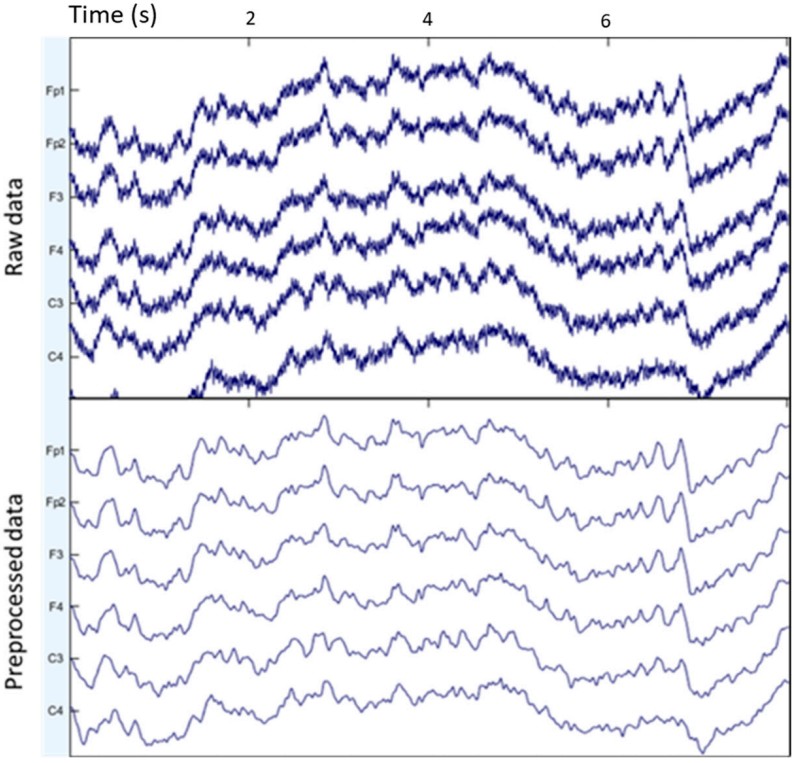

**Figure 2.** A snapshot of the same signal before and after being preprocessed.

### 3.3. Classification Benchmark

In order to benchmark the classification performance of the EEG dataset on the classification of AD vs. CN and FTD vs. CN, a variety of relatively simple feature extraction and classification techniques that can be easily reproduced and extended by other researchers were applied. While more complex algorithms (such as deep learning) and feature extraction techniques may provide better performance, the goal was to establish a basic benchmark for the dataset that could be easily validated and reproduced.

### 3.3.1. Feature Extraction

One of the most commonly extracted features for EEG classification tasks is the Relative Band Power (RBP) of the five frequency bands of interest of the brain activity. The five frequency bands are defined as [2]:

- Delta: 0.5–4 Hz
- Theta: 4–8 Hz
- Alpha: 8–13 Hz
- Beta: 13–25 Hz
- Gamma: 25–45 Hz

Moreover, according to the literature, AD patients exhibit changes in the RBP such as reduced alpha power and increased theta power.

In this paper, the EEG signals were first epoched to 4 s time windows with 50% overlap to create the population of the dataset that would be used for classification. Each epoch was labeled as AD, FTD, or CN.

In order to obtain the RBP, the Power Spectral Density (PSD) of the time-windowed signal for each frequency band was obtained using the Welch method [20], which splits the signal into overlapping segments and calculates each segment's squared magnitude of the discrete Fourier transform. A final estimate of the PSD is then created by averaging the obtained values. Finally, the relative ratio of PSD of each band for each epoch was calculated, resulting in the feature matrix that consisted of 5 features for each row. To calculate the relative ratio of PSD of a band, the PSD of the band is calculated and then divided by the PSD of the whole frequency range of interest, namely 0.5–45 Hz.

To illustrate the differences between the PSD of each group for each frequency band, Figure 3 is provided, which consists of heatmaps describing the PSD across the scalp, averaged across the AD, FTD, and CN groups.

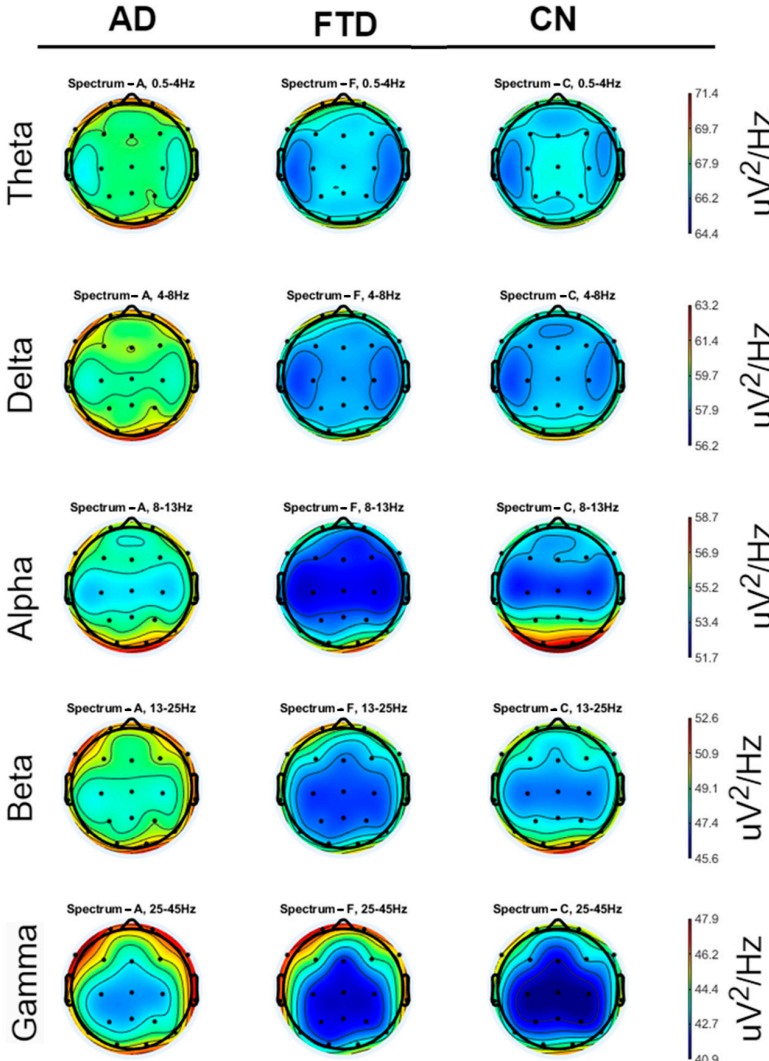

**Figure 3.** Scalp heatmaps of PSD across 5 frequency bands, averaged across AD, FTD, and CN groups.

3.3.2. Classification

The most used machine learning algorithms were used for the classification of AD-CN and FTD-CN to benchmark this dataset. The Leave-One-Subject-Out (LOSO) validation

method has been used for the performance evaluation of the algorithms [2]. In this validation methodology, all the epochs of one subject are left out as the test set while all the other epochs comprise the training set. This is repeated iteratively for every subject, and then the averaged performance metrics are calculated from the confusion matrix and presented. The performance metrics that were calculated were accuracy (ACC), sensitivity (SENS), specificity (SPEC), and the F1 score (F1). The machine learning algorithms used were LightGBM (hyperparameter optimized by Hyperopt [21]), Multilayer Perceptron (MLP) (1 hidden layer of 3 neurons), Random Forests, Support Vector Machine (SVM) (polynomial kernel), and kNN (k = 3). The results for the AD-CN classification are presented in Table 2 and the results for the FTD-CN classification are presented in Table 3.

**Table 2.** Classification Performance of Leave-One-Subject-Out validation for the AD-CN problem.

| AD/CN | ACC | SENS | SPEC | F1 |
|---|---|---|---|---|
| LightGBM | 76.43% | 76.01% | 76.16% | 76.12% |
| SVM | 73.14% | 71.89% | 75.98% | 73.74% |
| kNN | 71.23% | 69.67% | 74.19% | 72.81% |
| MLP | 73.12% | 73.00% | 74.63% | 74.82% |
| Random Forests | 77.01% | 78.32% | 80.94% | 75.31% |

**Table 3.** Classification Performance of Leave-One-Subject-Out validation for the FTD-CN problem.

| FTD/CN | ACC | SENS | SPEC | F1 |
|---|---|---|---|---|
| LightGBM | 72.43% | 61.13% | 80.74% | 67.32% |
| SVM | 70.14% | 62.41% | 75.98% | 68.32% |
| kNN | 67.34% | 59.67% | 76.13% | 70.81% |
| MLP | 73.12% | 63.00% | 78.63% | 72.82% |
| Random Forests | 72.01% | 72.32% | 80.94% | 66.31% |

## 4. User Notes

We encourage researchers to use the preprocessed data found in the derivatives folder. Moreover, when publishing a work based on this dataset, please check the "How to Acknowledge" section in the online dataset page and cite the appropriate article.

**Author Contributions:** Conceptualization, A.T.T. and P.A.; methodology, P.I., T.A. and N.G. (Nikolaos Grigoriadis); software, A.M., K.D.T. and N.G. (Nikolaos Giannakeas); validation, A.M., M.G.T. and K.D.T.; formal analysis, A.M. and M.G.T.; investigation, T.A., P.I. and N.G. (Nikolaos Grigoriadis); resources, P.I. and T.A.; data curation, A.M.; writing—original draft preparation, A.M.; writing—review and editing, D.G.T., N.G. (Nikolaos Giannakeas) and P.A.; visualization, A.M.; supervision, A.T.T. and D.G.T.; project administration, A.T.T.; funding acquisition, A.T.T. and E.G. All authors have read and agreed to the published version of the manuscript.

**Funding:** This research received no external funding.

**Institutional Review Board Statement:** This study was conducted in accordance with the Declaration of Helsinki and approved by the Scientific and Ethics Committee of AHEPA University Hospital, Aristotle University of Thessaloniki, under protocol number 142/12-04-2023.

**Informed Consent Statement:** Informed consent was obtained from all subjects involved in this study.

**Data Availability Statement:** The dataset is available at https://openneuro.org/datasets/ds004504 (accessed on 26 May 2023).

**Acknowledgments:** We acknowledge support of this work from the project "Immersive Virtual, Augmented and Mixed Reality Center of Epirus" (MIS 5047221), which is implemented under the action "Reinforcement of the Research and Innovation Infrastructure", funded by the Operational Program "Competitiveness, Entrepreneurship and Innovation" (NSRF 2014-2020), and co-financed by Greece and the European Union (European Regional Development Fund).

**Conflicts of Interest:** The authors declare no conflict of interest.

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
