# Peer review of "A Dataset of Scalp EEG Recordings of Alzheimer’s Disease, Frontotemporal Dementia and Healthy Subjects from Routine EEG"

_data, 2023_

Round 1

Reviewer 1 Report

Very good study and very well presented. 

Author Response

We thank the reviewer for accepting our study for publication.

Reviewer 2 Report

A dataset of scalp EEG recordings of Alzheimer’s disease, Frontotemporal Dementia and Healthy subjects from routine EEG

The paper describes an EEG dataset of Alzheimer's disease (AD), Frontotemporal dementia (FTD), and healthy control subjects. All data were recorded using the same device with 19 scalp electrodes. The raw and preprocessed data is provided as well as a method for classification is presented. One feature extraction method was used with five different classification methods and their results were provided using the leave-one-out approach.

The dataset is highly valuable for researchers and developers of EEG signal processing and analysis methods. Such data is difficult to obtain elsewhere. Recording of such data is typically not available outside of the clinical environment and even there limitations concerning GDPR often limit the data usage to the institution where it was acquired. As such, this work will highly benefit the research of AD and FTD.

The paper is well-written, easy to understand and, in my opinion, does not need any further modifications. I highly recommend accepting it for publishing.

Author Response

We thank the reviewer for their positive comments and for accepting our study for publication.

Reviewer 3 Report

The paper "A Dataset of Scalp EEG Recordings of Alzheimer’s Disease, Frontotemporal Dementia, and Healthy Subjects from Routine EEG" provides a comprehensive description of a valuable resting state EEG dataset comprising individuals with Alzheimer's disease, Frontotemporal dementia, and healthy controls. The dataset was meticulously collected using a clinical EEG system, incorporating 19 scalp electrodes, while participants were in a resting state with their eyes closed. One of the most notable aspects of this dataset is its potential for reuse, especially considering the increasing popularity of machine learning studies focused on Alzheimer's EEG analysis. Howerver, there are certain inconsistencies in the da

"A clinical EEG device (Nihon Kohden 2100) with 19 scalp electrodes (Fp1, Fp2, F7, F3, Fz, F4, F8, T3, C3, Cz, C4, T4, T5, P3, Pz, P4, T6, O1, O2) and 2 reference electrodes, placed on the mastoids for impedance check, was used for the recording of the EEG signals.""

Kindly include the names of the electrodes, specifically mentioning that they are placed on the mastoids (A1 and A2). Additionally, in the subsequent text, the authors should clarify that A1 and A2 are not only used for impedance check but also serve as reference for the EEG. 

Furthermore, please specify the location of the ground electrode used for common mode rejection.

"The recording montages were anterior-posterior bipolar and referential montage using Cz as the common reference."

I find the provided information to be contradictory. If the EEG channels are saved in a bipolar format, as stated in the initial part of the sentence, using either an anterior-posterior or a longitudinal montage, they cannot simultaneously be referenced to a single Cz common reference. Therefore, the possible options are either Fp1-F7, F7-T3, T3-T5, Fp1-F3, F3-C3, C3-P3, etc., or Fp1-Cz, F7-Cz, T3-Cz, Fp1-Cz, F3-Cz, C3-Cz, etc. It is crucial to clearly specify how the data was saved in the files.

"The signals were rereferenced to A1-A2 after applying a Butterworth band-pass filter with a frequency range of 0.5 to 45 Hz."

To enhance the clarity of the text, my understanding is that the EEG was initially acquired with Cz as the common reference. Subsequently, it was re-referenced in a bipolar anterior-posterior montage, and then the signals were further re-referenced to A1 and A2. I suggest considering the omission of the description regarding the bipolar referencing step, as it ultimately leads to the same results.

Furthermore, considering the authors' statement about referencing to A1 and A2, it raises a question about whether all 19 channels were referenced to the average value of A1 and A2, or if the left hemisphere electrodes were referenced to A1 and the right hemisphere electrodes to A2. Clarification regarding the specific referencing scheme employed would be helpful.

Figure 2, the time axis is missing. 

"Finally, the relative ratio of PSD of each band for each epoch was calculated, resulting in the feature matrix which consisted of 5 features for each row."

The dataset description lacks information on how the relative ratio of power spectral density (PSD) is calculated. My assumption is that the power of each of the five frequency bands is divided by the power from the 0.5-45 Hz range. However, it would be beneficial to explicitly state the method used for calculating the relative ratio of PSD to avoid any ambiguity.

Despite these inconsistencies, the dataset still holds significant potential for reuse, particularly given the rising interest in machine learning studies focused on Alzheimer's EEG analysis. The scarcity of publicly available EEG datasets further underscores the value of this contribution. 

Author Response

We thank the reviewer for their valuable comments.

Comment 1: Kindly include the names of the electrodes, specifically mentioning that they are placed on the mastoids (A1 and A2). Additionally, in the subsequent text, the authors should clarify that A1 and A2 are not only used for impedance check but also serve as reference for the EEG. 

Response: The sentence has been rephrased to “A clinical EEG device (Nihon Kohden 2100) with 19 scalp electrodes (Fp1, Fp2, F7, F3, Fz, F4, F8, T3, C3, Cz, C4, T4, T5, P3, Pz, P4, T6, O1, O2) and 2 electrodes (A1 and A2) placed on the mastoids for impedance check and as reference electrodes, was used for the recording of the EEG signals.”, so as that it is clarified that A1 and A2 electrodes are the reference electrodes and also used for impendance control. (Line 176-177)

Comment 2: Furthermore, please specify the location of the ground electrode used for common mode rejection. I find the provided information to be contradictory. If the EEG channels are saved in a bipolar format, as stated in the initial part of the sentence, using either an anterior-posterior or a longitudinal montage, they cannot simultaneously be referenced to a single Cz common reference. Therefore, the possible options are either Fp1-F7, F7-T3, T3-T5, Fp1-F3, F3-C3, C3-P3, etc., or Fp1-Cz, F7-Cz, T3-Cz, Fp1-Cz, F3-Cz, C3-Cz, etc. It is crucial to clearly specify how the data was saved in the files.

Response: We thank the reviewer for this comment. We have wrongfully mentioned that bipolar montage was used. The montage was referential (A1 A2) and the common mode rejection electrode was Cz. The statement has been restructured as: “The electrodes were placed according to the 10-20 international system. Each recording was performed according to the clinical protocol with participants being in a sitting position having their eyes closed. The recording montage was referential using Cz for common mode rejection.” (Line 180-182)

Comment 3: To enhance the clarity of the text, my understanding is that the EEG was initially acquired with Cz as the common reference. Subsequently, it was re-referenced in a bipolar anterior-posterior montage, and then the signals were further re-referenced to A1 and A2. I suggest considering the omission of the description regarding the bipolar referencing step, as it ultimately leads to the same results.

Response: We thank the reviewer for this comment. The protocol is as the reviewer described. We removed the description of the bipolar montage setting since no bipolar recordings were included in the dataset.

Comment 4: Furthermore, considering the authors' statement about referencing to A1 and A2, it raises a question about whether all 19 channels were referenced to the average value of A1 and A2, or if the left hemisphere electrodes were referenced to A1 and the right hemisphere electrodes to A2. Clarification regarding the specific referencing scheme employed would be helpful.

Response: We thank the reviewer for the valuable feedback. The average value of A1-A2 was used as reference for all electrodes. We made this clear by changing the sentence to: “The signals were re-referenced to the average value of A1-A2 after applying a Butterworth band-pass filter with a frequency range of 0.5 to 45 Hz.” (Line 190)

Comment 5: Figure 2, the time axis is missing. 

Response: Time axis has been added. The duration between each point is 2 seconds. The snapshot of the signal (raw and preprocessed) was taken at a random timestamp, so the points were numbered as 2,4,6 for illustration purposes.

Comment 6: The dataset description lacks information on how the relative ratio of power spectral density (PSD) is calculated. My assumption is that the power of each of the five frequency bands is divided by the power from the 0.5-45 Hz range. However, it would be beneficial to explicitly state the method used for calculating the relative ratio of PSD to avoid any ambiguity.

Response: We thank the reviewer for their comment. We added a sentence providing the information about how the relative PSD was calculated. To calculate the relative ratio of PSD of a band, the PSD of the band is calculated and then divided by the PSD of the whole frequency range of interest, namely 0.5-45 Hz.” (Line 230-232)

Reviewer 4 Report

The paper reports about the development and validation of an Electroencephalography (EEG) database made of signals acquired from patients affected by Alzheimer's disease, temporal dementia, and healthy controls during resting state. The paper is well written and the availability of such a database is relevant for the scientific community. In my opinion the Authors could insert some information about the statistical differences of the features considered for the classification (e.g., ANOVA and multiple comparison through t-test across the three groups)

Author Response

Comment 1: In my opinion the Authors could insert some information about the statistical differences of the features considered for the classification (e.g., ANOVA and multiple comparison through t-test across the three groups)

Response: We thank the reviewer for their valuable comment. Indeed, an illustrative comparison of the 3 different groups should be made. We have taken the initiative to extend our efforts and go above and beyond by adding Figure 3. This figure provides heatmaps of the PSD of each group for each frequency band. Now, the reader can observe the differences of the PSD not only across the different groups and bands, but also across the different electrodes.

Round 2

Reviewer 3 Report

Thank you. The Authors have addressed all of my concerns with the original manuscript. The revised manuscript is ready for publication.